# COVID-19 managed on respiratory wards and intensive care units: Results from the national COVID-19 outcome report in Wales from March 2020 to December 2021

**Simon M. Barry**[1,2]*, **Gareth R. Davies**[3], **Jonathan Underwood**[4,5], **Chris R. Davies**[6], **Keir E. Lewis**[3,7,8]

1 Department of Respiratory Medicine, Cardiff and Vale University Health Board, Cardiff, Wales, United Kingdom, 2 Respiratory Health Implementation Group, Cardiff, Wales, United Kingdom, 3 Respiratory Innovation Wales, Llanelli, Wales, United Kingdom, 4 Division of Infection and Immunity, Cardiff University, Cardiff, Wales, United Kingdom, 5 Department of Infectious Diseases, Cardiff and Vale University Health Board, Cardiff, Wales, United Kingdom, 6 Institute for Clinical Science and Technology, Cardiff, Wales, United Kingdom, 7 Department of Respiratory Medicine, Hywel Dda University Health Board, Carmarthen, Wales, United Kingdom, 8 School of Medicine, Swansea University, Swansea, Wales, United Kingdom

* simon.barry@wales.nhs.uk

**Data Availability Statement:** All relevant data are within the manuscript and its supporting files.

## Abstract

### Background

A COVID-19 hospital guideline was implemented across all 18 acute hospitals in Wales in March 2020, promoting ward management of COVID pneumonitis and data collected across the first 3 Waves of the pandemic (Wave 1 March 1st 2020 to November 1st 2020, Wave 2 November 2st 2020 to February 21st 2021 and Wave 3 June 1st 2021 to December 14th 2021). The aim of this paper is to compare outcomes for patients by admission setting and type of ventilatory support given, with a particular focus on CPAP therapy.

### Methods

This is a retrospective observational study of those aged over 18 admitted to hospital with community acquired COVID-19 between March 2020 and December 2021. The outcome of interest was in-hospital mortality. Univariate logistic regression models were used to compare crude outcomes across the waves. Multivariable logistic regression models were used to assess outcomes by different settings and treatments after adjusting for Wave, age, sex, co-morbidity and deprivation.

### Results

Of the 7,803 records collected, 5,887 (75.4%) met the inclusion criteria. Analysis of those cases identified statistically significant outcome improvements across the waves for all patients combined (Waves 1 to 3: 31.5% to 18.8%, p<0.01), all ward patients (28.9% to 17.7%, p<0.01), and all ICU patients (44.3% to 32.2%, p = 0.03). Sub group analyses identified outcome improvements in ward patients without any oxygen therapy (Waves 1 to 3:

**Funding:** Welsh Government Funds the Respiratory Health Implementation Group (RHIG) and Respiratory Innovation Wales. RHIG fund the Institute for Clinical Science and Technology (ICST), who create and implement the digital innovations. Welsh Government had no role in the study design, data collection and analysis, decision to publish or preparation of the manuscript.

**Competing interests:** The authors have declared that no competing interests exist.

22.2% to 12.7%, p<0.01), with oxygen therapy only (34.0% to 12.9%, p<0.01) and with CPAP only (63.5% to 39.2%, p<0.01). The outcome improvements for ICU patients receiving CPAP only (35.7% to 24.6%, p = 0.31) or invasive ventilation (61.6% to 54.6%, p = 0.43) were not statistically significant though the numbers being admitted to ICU were small. The logistic regression models identified important age and comorbidity effects on outcomes. The multivariable model that took these into account suggested no statistically significantly greater risk of death for those receiving CPAP on the ward compared to those receiving CPAP in ICU (OR 0.89, 95% CI: 0.49 to 1.60).

## Conclusions

There were successive reductions in mortality in inpatients over the three Waves reflecting new treatments and better management of complications. Mortality for those requiring CPAP was similar in respiratory wards and ICUs after adjusting for differences in their respective patient populations.

## Introduction

In March 2020, as the pandemic unfolded and hospital admissions rose across the UK, Wales launched its national COVID-19 hospital guideline. The guideline was disseminated through a digital implementation framework, Simple IMPlementation ScIence (SIMPSI) [1], deploying local facilitators to maximise guideline adoption, particularly targeting senior clinical decision makers (Consultants). The guideline was hosted on a digital platform requiring user registration with updates delivered in video format. The platform rapidly gained 4521 registrants including the vast majority of senior clinicians in frontline specialities delivering acute COVID care, achieving 170,000 views [2]. Given resource constraints, an early consensus decision was taken in Wales to manage patients with hypoxaemic COVID-19 pneumonitis on respiratory wards and avoid invasive ventilation where possible.

Welsh Government approved a national data collection tool, initially to collect data from Wave 1 but subsequently expanded to include Waves 2 and 3. Therefore, Wales developed an early strategy to try and standardise acute hospital care promoting ward management of COVID-19 pneumonitis and to systematically collect data on the outcomes of the COVID-19 pandemic at a national level.

This paper assesses the outcomes from community acquired COVID-19 in hospitalised patients across three distinct Waves of the pandemic. The impact of age, co-morbidities, and deprivation index on mortality were examined at each Wave, as emerging therapies, medical experience and vaccination affected outcomes over time. We focus on the outcomes of ward management of COVID-19 pneumonitis with oxygen alone and continuous positive airways pressure (CPAP) therapy and compare them with outcomes from the same respiratory support provided in ICUs in Wales.

## Methods

Retrospective observational study of data from adults (aged over 18 years) treated in hospital with COVID-19 for at least 24 hours in all 18 acute hospitals in Wales (median number of beds per hospital: 465, interquartile range 269–568).

## Study design

A digital tool collected anonymised patient and site-level outcomes (www.audit.clinicalscience.org.uk). This tool was endorsed by Welsh Government and the Information Governance Departments in each Health Board (HB). It was designed and implemented through the Institute for Clinical Science and Technology (ICST), who identified hospital audit and clinical leads to manage local data collection. The data collection tool was hosted within the National Pathway for Managing COVID-19 Infections in Secondary Care in Wales initiative (www.allwales.icst.org.uk).

Denominator data obtained from Public Health Wales (PHW) were all patients over 18 years admitted to hospital with a positive SARS-CoV-2 polymerase chain reaction (PCR) result between 1st March 2020 and 14th December 2021. Community acquired COVID-19 was defined as symptoms developing in the community and a positive PCR test in the community up to two weeks before admission, or within 48 hours of admission.

## Data collection

Demographic and clinical variables were collected for the index admission and inputted into the online tool. Mandatory fields included date of positive PCR swab, date and site of admission and discharge, age, gender, and outcome (death or discharge). Vaccination status was also a mandatory field for Wave 3. Fully vaccinated was defined as having had three vaccines. Supplementary fields included obesity, number of co-morbidities, frailty score and type and location of treatments given. Where patients received ICU and ward-based care or CPAP and 'oxygen alone', we allocated them to the highest-level-of-treatment group. The Welsh Index of Multiple Deprivation (WIMD) was derived from the patient's post-code and is the Welsh Government's official measure of relative deprivation for small areas in Wales [3]. We used the Welsh Governments preferred grouping of WIMD which is considered to better differentiate between levels of deprivation than quintiles. Available hospital notes were retrieved by record departments from the denominator lists and data inputted onto the online tool by junior doctors and audit teams.

Wave 1 was between March 1st 2020 and November 1st 2020

Wave 2 was between November 2st 2020 and February 21st 2021

Wave 3 was between June 1st 2021 and December 14th 2021

These 3 Waves were defined as they represent distinct spikes in the population prevalence of SARS COV-2 variants, relating predominantly to the wild type (wave 1), alpha (wave 2) and delta (wave 3) variants [4]. We stopped data collection in mid-December at the onset of the omicron wave.

## Outcomes

The primary outcome was in-hospital mortality, which was assessed across the Waves univariately by age, gender, sex, comorbidity, admission setting (ward or ICU), deprivation and modality of respiratory support. We also considered mortality by vaccination status for Wave 3.

## Missing data

Data were tabulated as the percentage completed for each parameter by HB in each Wave. HB represent geographical areas comprising a variable number of acute hospitals funded centrally

by Welsh Government to deliver integrated primary and secondary care. Mandatory fields achieved 100% completion but there was a large variation by HB for some of the non-mandatory fields (S1 Table). Data completion for WIMD was 95% since some post codes were missing or were from outside Wales.

No missing data imputation was undertaken for any variables or outcomes. BMI was rarely recorded in the notes and so levels of recorded obesity do not reflect true prevalence rates of 33% [5].

### Statistical methods

Baseline demographic and clinical characteristics and outcomes were combined across the HB (18 hospitals) and were grouped by Wave. Continuous variables are presented as median [inter-quartile range, IQR] and categorical variables as n (%), unless otherwise stated.

Crude outcomes were calculated as the number of deaths divided by the number of patients and presented as percentages. The confidence intervals for the crude outcomes were calculated using the Wilson method. The significance of crude outcome changes between waves was assessed by pairwise contrasts of waves following univariate logistic regression. Multivariable logistic regression models were used to assess the effect on mortality of each covariate after adjusting for all the other covariates, namely: Wave, total comorbidities, 10-year age band, gender and deprivation grouping. Two models were made and evaluated: a whole dataset model that used all records and included a ward or ICU admission covariate, and a subset model considering only those patients who received just CPAP, by ward or ICU. The value of introducing random effects via the inclusion of a hospital covariate was considered for both models.

The largest category in each covariate was used as the baseline category. No attempts were made to reduce either saturated models and no interaction terms were considered. No validation of the models on a subset or additional data set was undertaken.

The goodness of fit of both models were assessed by means of the Pearson Chi-square goodness of fit test, examination of the distributions of the predicted values and the Area under the ROC curve. Regression diagnostics were conducted by visual inspection of a range of residual plots for outliers. Outlying covariate patterns were assessed by the project team. All analyses were conducted in Stata BE 17.0.

### Ethics approval

Research Ethics Committee opinion was not required for this service evaluation in accordance with NHS guidance (https://www.hra-decisiontools.org.uk/research/docs/DefiningResearchTable_Oct2022.pdf).

## Results

Of the 7,803 records in the dataset, 179 were identified as duplicates and removed. After excluding those that didn't meet the case definition or had missing values in key fields, a total of 5,887 records remained, half of which were from Wave 2 (Fig 1).

### Whole cohort

The median age at admission dropped across the Waves for all admissions (3.5 years), all ward admissions (5 years) and ICU admissions (4 years). The median ages of patients admitted to ICU were around 10 years lower than for ward admissions for each Wave. The median number of comorbidities rose by one in Wave 3 though there was little difference between ward

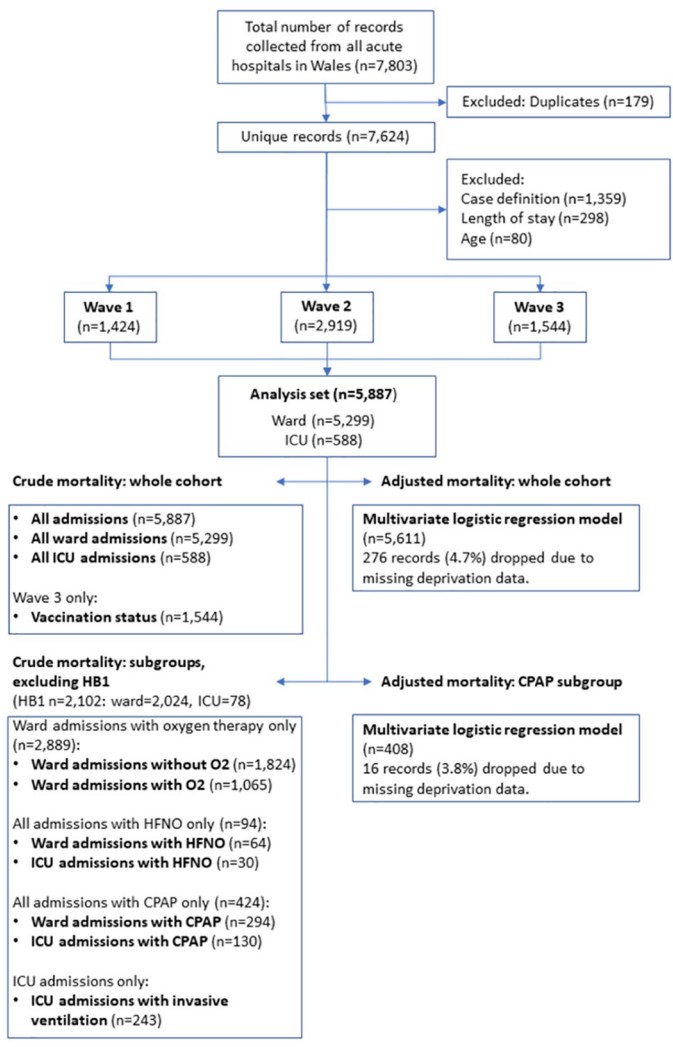

**Fig 1. Strobe diagram.**

and ICU patients. There was a consistent pattern of over representation of the population from the most deprived 30% of areas (around 41%), and under representation of the population from the least deprived 50% of areas (around 39%). There was a fairly consistent split between males (55%) and females (45%) across the waves for all admissions and ward admissions (Table 1). The sex split for ICU admissions was more marked being consistently around 65% males and 35% females across the waves (S2 Table). The length of stay for ward patients was a median of 7 days in Waves 1 and 2, but 5 days in Wave 3. For those on ITU, the median duration of length of stay was 15 days in Wave 1 and 13 days in Waves 2 and 3 (S3 Table and S1 Fig).

Fully vaccinated patients in Wave 3 were older and reported more comorbidities than the unvaccinated and partly vaccinated patients (S4 Table).

Across all waves combined, there were 1,398 deaths in the 5,887 admissions, an in-hospital mortality rate of 23.8%. The in-hospital mortality rates reduced considerably and statistically significantly across waves, for all admissions, ward admissions and ICU admissions. In-hospital mortality rates for ICU admissions remained consistently higher than for ward admissions

**Table 1. Whole cohort summary statistics.**

| | Wave | All admissions | | All ward admissions | | All ICU admissions | |
|---|---|---|---|---|---|---|---|
| | | Median | IQR | Median | IQR | Median | IQR |
| Age | 1 | 69.5 | 56.5 to 80 | 72 | 57 to 81 | 62 | 54 to 72 |
| | 2 | 69 | 55 to 80 | 71 | 56 to 81 | 61 | 52 to 68 |
| | 3 | 66 | 49 to 78 | 67 | 50 to 79 | 58 | 48 to 68 |
| | All | 69 | 54 to 79 | 70 | 55 to 80 | 61 | 52 to 69 |
| Comorbidities | 1 | 2 | 1 to 4 | 2 | 1 to 4 | 2 | 1 to 3 |
| | 2 | 2 | 1 to 4 | 2 | 1 to 4 | 2 | 1 to 4 |
| | 3 | 3 | 1 to 4 | 3 | 1 to 4 | 2 | 1 to 4 |
| | All | 2 | 1 to 4 | 2 | 1 to 4 | 2 | 1 to 3 |
| | | 30% most | 50% least | 30% most | 50% least | 30% most | 50% least |
| Deprivation (% from areas in most deprived 30% and least deprived 50%) | 1 | 41.0 | 39.2 | 40.1 | 39.4 | 45.7 | 38.5 |
| | 2 | 41.3 | 39.2 | 41.8 | 38.8 | 36.4 | 43.6 |
| | 3 | 41.5 | 37.2 | 41.4 | 37.1 | 41.7 | 38.3 |
| | All | 41.3 | 38.7 | 41.3 | 38.5 | 41.2 | 40.4 |
| | | Male | Female | Male | Female | Male | Female |
| Sex (%) | 1 | 56.9 | 43.1 | 55.3 | 44.7 | 65.1 | 34.9 |
| | 2 | 54.3 | 45.7 | 53.3 | 46.7 | 66.0 | 34.0 |
| | 3 | 54.7 | 45.3 | 53.9 | 46.1 | 63.6 | 36.4 |
| | All | 55.0 | 45.0 | 53.9 | 46.1 | 65.1 | 34.9 |

across the waves. There was a statistically significant difference (p<0.01) in the Wave 3 in-hospital mortality rates for those fully vaccinated and those unvaccinated (S5 Table).

Univariate logistic regression analyses of the whole cohort indicated higher odds of in-hospital mortality for ICU patients than ward patients (OR 2.42, 95% CI 2.02 to 2.90), reducing mortality across the Waves, increasing mortality with greater levels of comorbidity and with increasing age, lower mortality for females and no deprivation effect (S6 Table and S2 Fig).

The multivariable logistic regression model for the whole cohort indicated very similar effects to the univariate analyses. One notable difference was the increase in the odds ratio for ICU patients compared to ward patients (OR 4.41, 3.54 to 5.50) after adjusting for the other covariates (Fig 2 and S7 Table).

The Pearson chi-square goodness of fit test indicated no problems with the fit of the model (p = 0.25). The area under the Receiver Operating Characteristic (ROC) curve suggested the model had satisfactory discriminatory powers (0.77). Examination of the residual plots highlighted a number of outlying covariate patterns. On inspection, none of these appeared to be untoward and all data were left in the model (S1 and S2 Appendices).

The addition of hospital site as a random effects variable made little difference to the model output values and had no effect on the patterns and interpretations. For ease of interpretation the random effects variable was not included in the final model.

## Treatment subgroups

Age at admission dropped slightly across the Waves for ward patients who received no oxygen treatments at all or received oxygen therapy only (S8 Table). These two groups were very similar for each wave in terms of median age and number of comorbidities and both exhibited over representation of the most deprived areas and under representation of the least deprived areas (S8 Table). A greater percent of males had oxygen therapy than females across all waves,

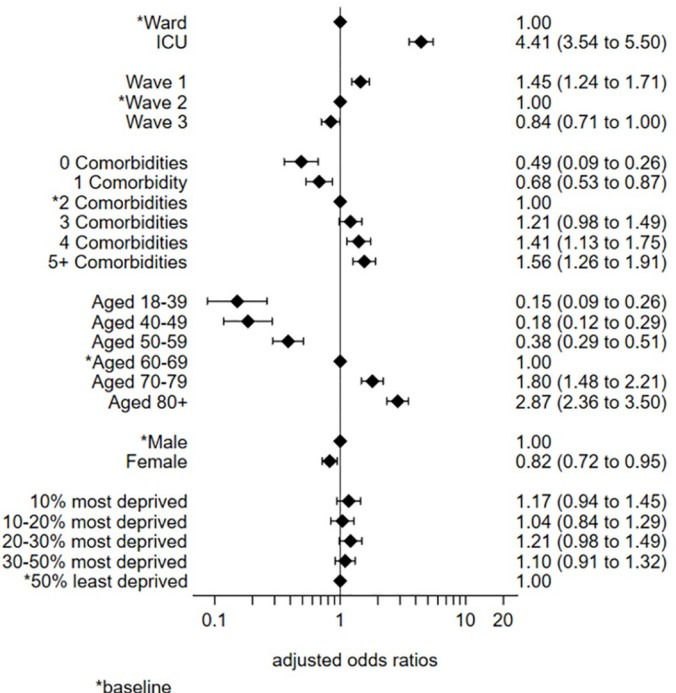

**Fig 2. Multivariable logistic regression model: Whole cohort.**

though for both sexes the percent rose from Wave 1 to Wave 2 then fell from Wave 2 to Wave 3 (S9 Table).

No patients received only HFNO in Wave 1 and the numbers in Waves 2 & 3 were small: ward 35 & 29, ICU 15 & 15. Median age increased from 65 to 71 for ward HFNO patients and from 52 to 55 for ICU HFNO patients from Wave 2 to Wave 3, with ward patients being considerably older than ICU patients. The ward HFNO patients reported slightly higher comorbidities and both groups exhibited the deprivation effect seen in the other sub groups. Females who received HFNO only were very slightly more likely to be treated in ICU than males (S10 and S11 Tables).

Age at admission dropped by five years across the Waves for ward patients who received CPAP, though barely changed for patients receiving CPAP in ICU. Ward CPAP patients were considerably older than ICU CPAP patients, by 15.5 years in Wave 1 and 10 years in Waves 2 and 3. The ward CPAP patients had slightly higher comorbidity and both groups exhibited the deprivation effect seen in the other sub groups, though to a slightly lesser extent of around 7.5% (Table 2). The male/female split was similar in both settings with around 60% of CPAP patients being male (S12 Table). The time spent on CPAP for those on wards (8–10 days) was similar to those receiving this care in ICU (10–11 days) across all three waves (S3 Table).

Age at admission dropped from 62 to 60 to 58 for patients who were invasively ventilated. Comorbidity levels were slightly lower for invasively ventilated patients than for CPAP or HFNO patients and were similar to ward patients with or without oxygen therapy. The deprivation effect seen in the other sub groups was also apparent for this group, with a greater under representation of the least deprived 50% of around 15%. There was a marked difference in the male/female split in both settings with around 70% of invasively ventilated patients being male (S12 and S13 Tables).

**Table 2. CPAP subgroup summary statistics.**

| | Wave | Ward CPAP admissions | | ICU CPAP admissions | |
|---|---|---|---|---|---|
| | | Median | IQR | Median | IQR |
| Age | 1 | 71 | 61 to 77 | 55.5 | 46 to 64 |
| | 2 | 69 | 57.5 to 77 | 59 | 51 to 65 |
| | 3 | 66 | 55 to 76 | 56 | 50 to 66 |
| | All | 69 | 57 to 77 | 57.5 | 50 to 65 |
| Comorbidities | 1 | 2 | 2 to 3 | 1 | 0 to 3 |
| | 2 | 3 | 2 to 4 | 2 | 1 to 4 |
| | 3 | 3 | 2 to 5 | 2 | 1 to 3 |
| | All | 3 | 2 to 4 | 2 | 1 to 4 |
| | | 30% most | 50% least | 30% most | 50% least |
| Deprivation (% from areas in most deprived 30% and least deprived 50%) | 1 | 44.3 | 26.2 | 45.5 | 36.4 |
| | 2 | 37.0 | 44.5 | 30.1 | 53.4 |
| | 3 | 32.1 | 51.3 | 41.3 | 41.0 |
| | All | 37.2 | 42.5 | 35.0 | 48.0 |
| | | Male | Female | Male | Female |
| Sex (%) | 1 | 66.7 | 33.3 | 57.1 | 42.9 |
| | 2 | 56.6 | 43.4 | 58.7 | 41.3 |
| | 3 | 59.5 | 40.5 | 65.9 | 34.1 |
| | All | 59.5 | 40.5 | 60.8 | 39.2 |

The in-hospital mortality rates reduced considerably and statistically significantly across waves, for ward admissions without any oxygen therapy, ward admissions with oxygen therapy only, and ward admissions with CPAP (Fig 3). The in-hospital mortality rates reduced for ICU admissions with HFNO, ICU admissions with CPAP and ICU admissions with invasive ventilation but not statistically significantly so. The in-hospital mortality rates increased slightly for ward patients with HFNO though not statistically significantly (S14 Table).

Univariate logistic regression analyses of the CPAP subgroup indicated much lower odds of in-hospital mortality for ICU CPAP patients than for ward CPAP patients (OR 0.35, 95% CI 0.22 to 0.55), reducing mortality from Wave 1 to Wave 2 only, increasing mortality with

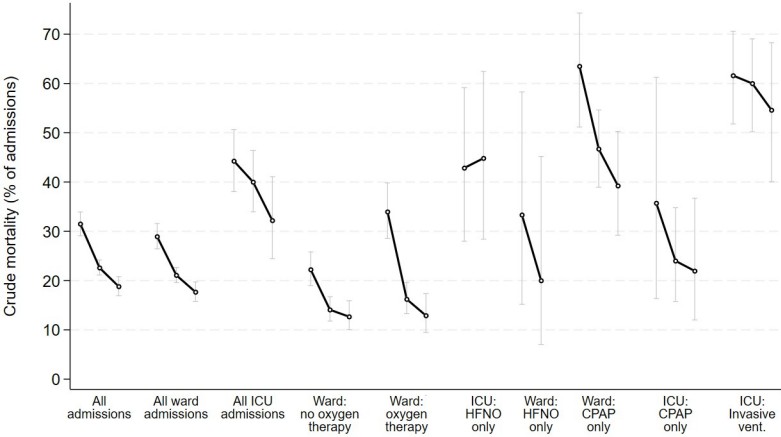

**Fig 3. Crude outcomes: Whole cohort and treatment subgroups with 95% confidence intervals.**

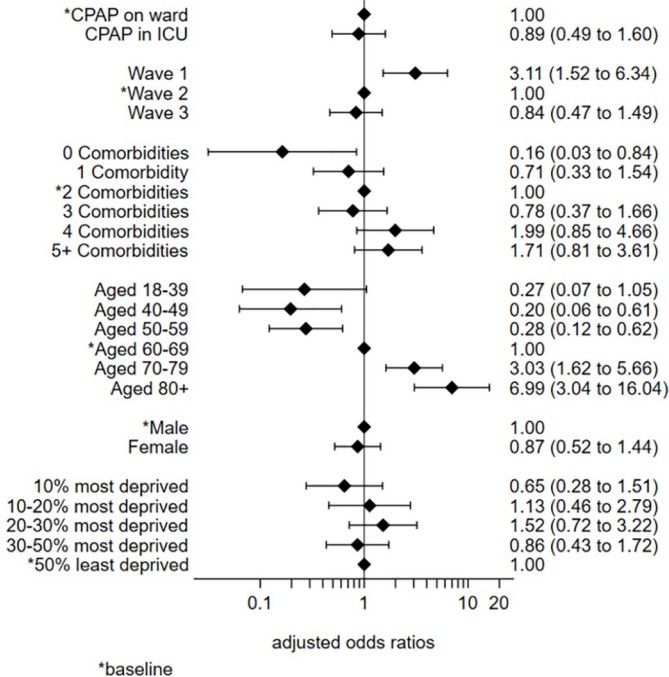

**Fig 4. Multivariate logistic regression model: CPAP only.**

greater levels of comorbidity and with increasing age, no difference between the sexes and no deprivation effect (S15 Table and S3 Fig).

The multivariable logistic regression model of the CPAP cohort indicated very similar effects to the univariate analyses. One notable difference was the change to equivalence in the odds ratio for ICU CPAP patients compared to ward CPAP patients (OR 0.89, 0.49 to 1.60) after adjusting for the other covariates (Fig 4 and S16 Table).

The Pearson chi-square goodness of fit test indicated no problems with the fit of the model (p = 0.30). The area under the Receiver Operating Characteristic (ROC) curve suggested the model had excellent discriminatory powers (0.84). Examination of the residual plots highlighted a number of outlying covariate patterns. On inspection, none of these appeared to be untoward and all data were left in the model (S3 and S4 Appendices).

The addition of hospital site as a random effects variable made little difference to the model output values and had no effect on the patterns and interpretations. For ease of interpretation the random effects variable was not included in the final model.

## Discussion

This study from all acute hospitals in Wales demonstrated reductions in mortality over successive waves of the pandemic. After adjusting for confounding variables we found no statistically significant difference in mortality between those that received CPAP on wards or in ICU. The data on 5887 admissions represents approximately 40% of total community acquired COVID-19 admissions in acute hospitals in Wales. Our overall mortality rate of 23.8% was consistent with other studies [6–8], but higher than the 5% reported in Japan [9]. This latter study comprised a younger, less severe cohort with only 26% requiring oxygen and 2.8% HFNO/CPAP, compared to 64% on oxygen and 17.6% on CPAP/HFNO on wards in Wales. Our patients had

higher mortality associated with increasing co-morbidities, age and male sex, consistent with previous studies [6–9].

The mortality rates for those in ICU and anyone on CPAP/HFNO were higher than those with oxygen alone at each Wave, indicating a group with more severe pneumonitis. Our longitudinal analyses found mortality improvements for each treatment group across successive Waves. The reduction in all-cause mortality over time was likely due to improved experience and effects of new treatments and it is notable that the biggest reduction was between Wave 1 and Wave 2 where the introduction of dexamethasone became the standard of care for anyone requiring oxygen [10]. There were improvements in other ward-based treatments between Waves 1–2 including more extensive use of proning, a lower threshold for testing for pulmonary emboli and awareness of complications such as pneumomediastinum [11] and super-added infections. By end of Wave 2 there were further impacts of IL6-inhibitors [12] and more experience with CPAP and by Wave 3, vaccines [13, 14] and anti-COVID antibody therapies [15] were impacting on the severity of illness (Fig 5).

We found that those the 'most deprived 30%' were over-represented, accounting for 42% of hospital admissions with COVID-19. There was an improvement in mortality for all levels of deprivation with each successive Wave suggesting all patients had similar access to improvements in hospital medical care over time. Multivariate analysis indicated that deprivation was not an independent risk factor for mortality overall. This finding is at variance with other data [16, 17], but our analysis is only of hospitalised community acquired COVID-19 in adults so cannot reflect population level associations between COVID mortality and deprivation. Indeed, Office for National Statistics (ONS) data demonstrated a clear association between deprivation and mortality in England and Wales [18].

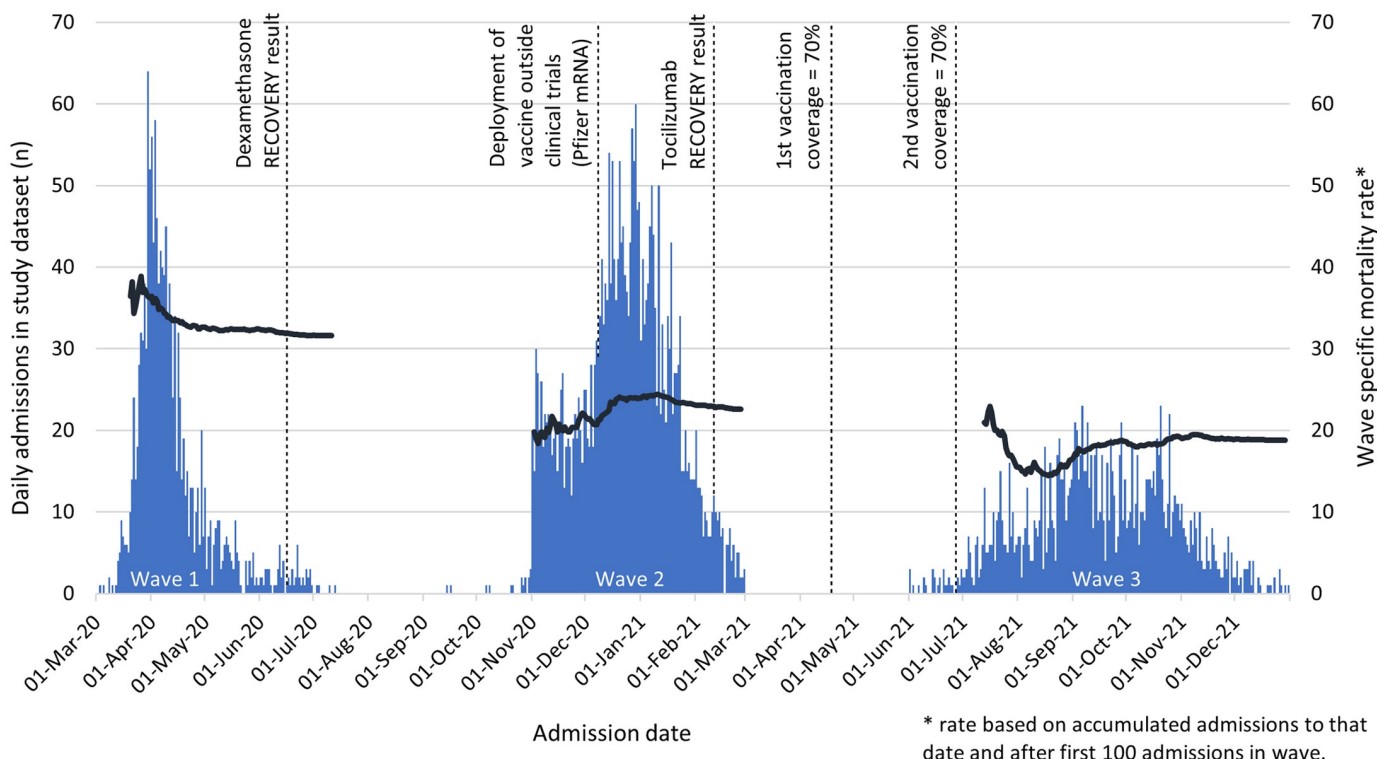

**Fig 5. COVID-19 admissions, wave-specific mortality and timeline of key treatments and vaccination.**

Unvaccinated patients represented 77% of admissions during Wave 3 and the unvaccinated had a mortality rate of 20.5% compared to 4.8% in those fully vaccinated. These findings support the strong protective effects of vaccines against hospitalisation and death.

At the beginning of the pandemic, key individuals in the respiratory and intensive care community in Wales agreed that patients should be managed on respiratory wards and given oxygen, with the addition of CPAP or HFNO according to the guideline and only those failing on this treatment who were suitable for escalation, or who initially presented with severe hypoxaemia should be managed in critical care. This decision was taken partly in order to protect limited ICU beds as Wales has a lower number per capita than England and about half that of the European average [19], but also because there was a consensus view early on that invasive ventilation was best avoided. It is of interest therefore that UK wide ICU data collection demonstrated that Wales had a lower proportion of patients managed in intensive care on basic respiratory support (oxygen, CPAP and HFNO) compared to the UK as a whole (16.4% compared to 25.7% for the first wave and 27.3% compared to 42.8% for the second wave) [20]. For the first Wave, CPAP was the preferred modality as there were concerns about oxygen supply in some hospitals with HFNO. As experienced increased with CPAP in Waves two and three, HFNO was increasingly used to provide breaks from CPAP and in those unable to tolerate CPAP. For those on wards using either treatment modality, the majority (83%) had CPAP.

Our study provides further support to the approach of treating COVID-19 pneumonitis with CPAP [21] and our findings reflect the reality of older, more comorbid patients being managed in ward environments. It is reassuring to note that once confounding factors were adjusted for, there was no significant difference in mortality for patients receiving CPAP on wards or in ICU. Age was an important factor for survival with CPAP with those aged over 80 having an increased odds ratio for death of 7 compared to those with an age of 60–69.

There are several strengths to our study: First, this was a truly national dataset from all acute hospitals over three distinct waves of the pandemic. Second, due to the online data collection tool, there were complete datasets for mandatory fields so that in-hospital mortality rates, length of stay and deprivation were reliably measured. Third, we disseminated the guideline widely to healthcare professionals using a digital system and promoted ward-based care including CPAP and HFNO. Thus, our study provides evidence on the outcomes of ward management of hypoxaemic patients with COVID-19 pneumonitis.

This study suffers from the limitations of retrospective case-note reviews with missing data on non-mandatory fields including frailty scores and treatments particularly from one HB (S1 Table). Obesity, a recognised risk factor for COVID [22] was also poorly recorded. Our prerequisite for symptoms and positive swab within 48 hours of admission would have missed a proportion of community acquired cases that could have had a positive PCR up to 5–7 days after admission, but we have argued this approach could have included nosocomial cases which we have previously shown are a frailer subgroup with worse outcomes [23]. We have only measured mortality from the index admission and are uncertain of outcomes following discharge from hospital. We do not have data relating to severity of co-morbidities, we did not measure re-admissions and we have not mapped the outcomes against variants of the virus, but the Waves largely corresponded to the Wuhan wild type, alpha and delta waves respectively. Finally, whilst we have collected data on a large proportion of admissions across all acute Hospitals in Wales, we cannot exclude case ascertainment bias according to notes availability although with over 5800 records, we feel this is unlikely.

We provide national data from 18 hospitals over 21 months, following the implementation of a hospital guideline supporting the ward management of patients with COVID-19 pneumonitis. We report declining mortality rates with each successive wave highlighting the beneficial

effects of new treatments and vaccination. We demonstrate that outcomes with CPAP and oxygen delivered in respiratory wards were similar to those in ICU, thus supporting ward management of COVID-19 pneumonitis. Our study also provides insights on implementing digital guidelines for future pandemics.

## Supporting information

**S1 Fig. Length of stay, whole cohort, and treatment subgroups.**
(PDF)

**S2 Fig. Univariate logistic regression odds ratios, whole cohort.**
(PDF)

**S3 Fig. Univariate logistic regression odds ratios, CPAP subgroup.**
(PDF)

**S1 Table. Non-mandatory data completeness, whole cohort.**
(PDF)

**S2 Table. Whole cohort counts and percents.**
(PDF)

**S3 Table. Length of stay, whole cohort and treatment subgroups.**
(PDF)

**S4 Table. Wave 3 vaccination status counts and percents.**
(PDF)

**S5 Table. Whole cohort crude outcomes.**
(PDF)

**S6 Table. Univariate logistic regression model, whole cohort.**
(PDF)

**S7 Table. Multivariable logistic regression model, whole cohort.**
(PDF)

**S8 Table. Subgroup summary statistics, oxygen therapy.**
(PDF)

**S9 Table. Subgroup counts and percents, oxygen therapy.**
(PDF)

**S10 Table. Subgroup summary statistics, HFNO only.**
(PDF)

**S11 Table. Subgroup counts and percents, HFNO only.**
(PDF)

**S12 Table. Subgroup counts and percents, CPAP and invasive ventilation.**
(PDF)

**S13 Table. Subgroup summary statistics, invasive ventilation.**
(PDF)

**S14 Table. CPAP subgroup crude outcomes.**
(PDF)

**S15 Table. Univariate logistic regression model, CPAP subgroup.**
(PDF)

**S16 Table. Multivariable logistic regression model, CPAP subgroup.**
(PDF)

**S1 Appendix. Multivariable logistic regression model, whole cohort, goodness of fit.**
(PDF)

**S2 Appendix. Multivariable logistic regression model, whole cohort, regression diagnostics.**
(PDF)

**S3 Appendix. Multivariable logistic regression model, CPAP subgroup, goodness of fit.**
(PDF)

**S4 Appendix. Multivariable logistic regression model, CPAP subgroup, regression diagnostics.**
(PDF)

## Acknowledgments

Public Health Wales for providing the denominator data on COVID admissions, Hannah Sharp (Institute Clinical Science and Technology) and the local facilitators; Sarah Bowen, Carol Llewellyn-Jones, Abdelrahman Mohamed, Inder Singh, Shehnoor Tarique, Carla Dos Santos Gil, Sara Fairbairn, Tarek Dihan, Rhodri Edwards, Matt Brouns, Ramsey Sabit, Amit Benjamin, Jacqueline Woolley, Claire Kilduff, Daniel Menzies, Liz Brohan, Elen Rowlands, Alexandra Scott, Sinan Eccles, Anna Lewis, Sharon Ragbetli, Sion O'Keefe, Joanne Stimpson, Laura Gingell, Liz Evans, Ian Bebb, Jane Christmas and Favas Thaivalappil. All junior doctors and audit teams who helped input data.

## Author Contributions

**Conceptualization:** Simon M. Barry, Keir E. Lewis.

**Formal analysis:** Gareth R. Davies.

**Funding acquisition:** Simon M. Barry.

**Methodology:** Gareth R. Davies, Jonathan Underwood.

**Project administration:** Keir E. Lewis.

**Resources:** Chris R. Davies.

**Software:** Chris R. Davies.

**Supervision:** Simon M. Barry, Jonathan Underwood.

**Visualization:** Chris R. Davies.

**Writing – original draft:** Simon M. Barry.

**Writing – review & editing:** Gareth R. Davies, Jonathan Underwood, Chris R. Davies, Keir E. Lewis.

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
