## [Decision Letter · Decision Letter 0]

9 May 2023

PONE-D-23-05105COVID-19 Managed on Respiratory Wards and Intensive Care Units: Results from the National COVID-19 Outcome Report in Wales From March 2020 to December 2021.PLOS ONE

Dear Dr. Barry,

Thank you for submitting your manuscript to PLOS ONE. After careful consideration following review by three external reviewers, we feel that it has merit but does not fully meet PLOS ONE’s publication criteria as it currently stands. Therefore, we invite you to submit a major revised version of the manuscript that addresses the points raised during the review process.

Please read and respond to all of the peer review comments. Please note that your revision may be subject to further review and that this initial decision does not guarantee acceptance.

We look forward to receiving your revised manuscript.

Kind regards,

Anne K. Örtqvist Rosin, Ph.D, M.D

Academic Editor

PLOS ONE

Journal Requirements:

"Welsh Government Funds the Respiratory Health Implementation Group (RHIG) and Respiratory Innovation Wales. RHIG fund the Institute for Clinical Science and Technology (ICST), who create and implement the digital innovations."           

“no authors have competing interests”

Please complete your Competing Interests on the online submission form to state any Competing Interests. If you have no competing interests, please state "The authors have declared that no competing interests exist.", as detailed online in our guide for authors at http://journals.plos.org/plosone/s/submit-now.  

Reviewers' comments:

Reviewer's Responses to Questions

**Comments to the Author**

1. Is the manuscript technically sound, and do the data support the conclusions?

Reviewer #1: Yes

Reviewer #2: Yes

Reviewer #3: Partly

2. Has the statistical analysis been performed appropriately and rigorously? 

Reviewer #1: Yes

Reviewer #2: Yes

Reviewer #3: No

3. Have the authors made all data underlying the findings in their manuscript fully available?

Reviewer #1: Yes

Reviewer #2: Yes

Reviewer #3: Yes

4. Is the manuscript presented in an intelligible fashion and written in standard English?

Reviewer #1: Yes

Reviewer #2: Yes

Reviewer #3: Yes

5. Review Comments to the Author

Reviewer #1: Dear authors,

Thank you for the opportunity to review this nationwide, retrospective analysis of hospitalized patients with Covid-19.

Data was collected though an online, anonymised database and the project was assessed by an ethical review board but no opinion was required due to the nature of the study.

Title: Reflects to contents of the study

Abstract: Straigth to the point. Reads well. Aim, findings and conclusion aligned.

Introduction:

LL43-50: The authors make a huge point of the national guideline. This part is quite unexpected and please justify why this part is required, as it does not seem to be relevant to the data presented. Or was the guideline and database tighly connected? Please clarify.

L50: The last meaning on avoiding invasive mechanical ventilation whenever possible spikes an interest. This strategy is similar to what was experienced from northern Italy. In light of your results, I think your viewpoint regarding this practice would be quite interesting. Did the strategy change over time when pressure on ICU/ventilator beds decreased?

Method:

L967: Please specify when mortiality was assessed. During index admission? Hospital mortality? 30d? If patients were discharged to for example a nursing facility and succumbed shortly thereafter, was this registered as an primary event outcome? Please specify.

Statistical analysis: Appears to be sound.

Results:

Well presented, Fig 1 and Fig 2 are informative and yields relevant information.

Just out of curiosity, why did you chose the specific baseline/reference categories? Eg Wave 2 as reference instead of 1 or 3? (No need to revise!)

Discussion:

LL216-218: Consider omitting this, the aim was already stated. I would rather see your major findings first in the discussion.

LL227-228: I would add that the reduction in mortality over was was likely due to.... There is no data to support increasing experience - for pharmaceuticals, pls refer to the supplement table.

LL230-233: Does your data support this?

LL236-240: I think this is a v imp finding - consider adding that this could be attributed to the publicly funded healthcare system (if you agree of course!)

LL255-265: Here you pick up on the topic I commented on earlier - I would like to see a deeper analysis. In hindsight, would you do the same choice again? This is important for future desicion-making! Did the propotion of patients on invasive ventilation change over time?

LL282: Consider adding that less than half the hospitalised patients were registered to the database.

I enjoyed reading this report and I think it is valuable for future decision-making and hoping that my feedback could help improving the report further.

Reviewer #2: Thank you for the opportunity to review this interesting work. Data are limited on the association between sociodemographic status and outcomes as well as the comparison of outcomes by wards and receipt of respiratory care.

Please find below suggested comments for consideration:

1. Data are plural and the text should be updated accordingly (e.g. line 20 “data were collected” rather than “data was collected”)

2. Abstract, methods – line 23, please define adults (> 18 years).

3. Abstract, methods – suggest including a sentence regarding the statistical analyses performed.

4. Abstract, results – suggest including the number of hospitals

5. Ensure that “COVID-19” is used throughout the text rather than “COVID” (e.g. lines 50, 58 etc.)

6. Methods, line 65 – suggest including additional information to describe the acute care hospitals if available (e.g. average bed size, adult vs. pediatric sites, teaching vs. non-teaching hospitals)

7. Methods, line 69 – please spell out HB the first time it is referenced

8. Methods, line 76 “continuous notes review by local clinical teams” – suggest specifying or describing what this means. Does it mean that admissions records were checked to see which ones had clinician notes available? Please clarify.

9. Methods – suggest spelling out CPAP and defining HFNO which is referenced in the results but not in the methods

10. Methods, lines 85-87 – suggest providing additional details regarding WIMD, specifically defining the levels or categories of deprivation status (e.g. 50% most deprived).

11. Methods, line 96 – please define mortality. It is all-cause in-hospital mortality? 30-day mortality?

12. Methods, lines 102-103 – if possible, suggest indicating why some post codes did not link to a deprivation status (e.g. outside of geographic area of study? Or not mapped to status?)

13. Methods, line 105 – if possible, please provide further information regarding how missing data imputation was used.

14. Methods, line 105-106 – to confirm was BMI excluded from the analysis due to incomplete records? Please clarify

15. Methods, line 109 – Health Boards is spelled out here. For consistency, suggest spelling out the first time it is used than consistently using HB.

16. Results, line 133 – lower case “wave”, for consistency suggest using capital “Wave”.

17. Results, lines 133-135 – the term patients, datasets and patient dataset are used interchangeably here. Suggest selecting one term to use consistently.

18. Results, lines 138-139 – is this referencing the median or average age? Please clarify. And “those admitted in Wave 3 were 4 years younger” – the median or average age was 4 years younger?

19. Results, lines 139-140 “likely reflecting increased vaccination rates in the older age groups protecting against admission” – this is not a result but better suited in the discussion. Also, are there data to support this statement? Are population level age-stratified vaccination rates available?

20. Table 1 – consider adding p value results.

21. Results, lines 152-159 – suggest to include key relevant data (e.g. odds ratios and 95% confidence intervals) either in the text or figures.

22. Results, lines 167-175 – as above, helpful to have odds ratios and 95% CI reported here for key results.

23. Results, lines 167-175 – suggest reporting as “odds of death” (e.g. the effect of age persisted with a lower odds of death at lower age…”)

24. Results, lines 193-195 – suggest confirming that this difference is statistically significant? Regardless, is it clinically significant? A difference of 1 comorbidity?

25. Results, lines 208-209 – suggest adding definition of fully vaccinated in methods. Also, suggest clarifying sentence to “The overall mortality among unvaccinated patients was 20.5%, compared to 4.8% among patients who were fully vaccinated.”

26. Results, lines 211-212 – suggest comparing to the same group (e.g. 81% were unvaccinated compared to 26.6% unvaccinated). The comparison of unvaccinated to vaccinated is not useful.

27. Discussion, line 219 – the data represents approximated 40% of total CA COVID admissions in acute hospitals in Wales? Of do the data represent 40% of all admissions (COVID and non-COVID)? Perhaps clarify because above it is stated that every acute hospital in Wales is captured in these data.

28. Discussion, line 223 – suggested updating to “our patients had higher mortality associated with increasing…”

29. Discussion, line 247 – suggest using consistent terminology “fully vaccinated” vs. “double vaccinated with a booster”

30. Discussion, line 247 – hesitant to use the word “confirm” here as this study did not control for other factors which may have either contributed to death (e.g. underlying comorbidities, age etc.) or improved chance of survival (such as age or prior/natural immunity)

31. Discussion, lines 247-248 – cannot conclude that vaccines protect against hospitalization as this was not measured in this study (you would need community level data to compare to those CA COVID-19 positive not hospitalized by vaccination status).

32. Discussion, lines 266 – consider conducting a multivariable logistic regression to compare those who received CPAP and those who did not with the outcome of mortality to provide further support/strength this conclusion.

33. Figure 1 – should the grey bar be “non ICU admissions”?

Reviewer #3: Re: PONE-D-23-05105

COVID-19 Managed on Respiratory Wards and Intensive Care Units: Results from the National COVID-19 Outcome Report in Wales From March 2020 to December 2021

Thank you for asking me to review this retrospective study of 5887 patients with COVID-19 admitted to all 18 acute hospitals in Wales between March 2020 and December 2021. The authors report declining mortality for those requiring oxygen or CPAP over the pandemic period with stepwise reduction over successive waves from 1 to 3. They also report no difference in mortality outcomes between those who received CPAP in wards as opposed to ICU.

The attempt to standardise management across Welsh hospitals and collect data about this was laudable. There is a lot of potentially interesting data here but the presentation of the results is disorganised. The paper lacks a clear aim. The paper falls between a scientific paper aiming to address a specific research question and a generic health department style report covering multiple aspects of care provided to COVID patients. If the aim is the former, it should be structured to address a specific question (e.g. change in mortality over the 3 waves). If the aim is the latter, it should be structured as a report with more complete ordered information with less commentary, although this then might be difficult to meet the requirements for publication as a scientific paper, and would be better suited to an on-line health department report.

The authors have also focused in on a result from a subgroup (CPAP in ICU) which does not appear to be a primary aim of the paper. If the main aim is to determine the how outcomes differ between patients who are treated with CPAP in the ward as opposed to ICU, then there should be a much more robust analysis to back up their findings, and this should become the focus of the paper. (See later comment also).

Outcomes from respiratory wards and ICUs are reported. Does this mean that patients admitted to other wards are not included? What happened to COVID patients put on other wards or have all patients admitted with COVID been included in the study?

Do the authors have information on how many patients received invasive ventilation? This is vital to interpret the prevalence and outcomes associated with provision of other respiratory / ventilatory / oxygen therapies.

Specific comments

Abstract

Timing of COVID waves differed in many countries. Summary timings for each should also be provided in abstract.

Introduction

Paragraph about report on hospital acquired COVID not relevant? Intro should describe the knowledge gap to be addressed by this paper.

The aim of the paper should be more clearly spelt out.

Methods

Mortality – clarify ‘in-hospital mortality’, ‘x-day mortality’, ‘case fatality’ etc

Missing data – More information should be provided about how imputation was done. What was imputed and how?

How many were missing mortality outcome data? These should just be excluded and reported as missing, not undergo imputation of outcomes.

The setting should be more clearly described. This is important for an international audience. Authors state data collection occurred at all 18 hospitals but report 6 health boards. For non-UK readers such as myself, it would help to explain what an HB is. I presume this is an organisational grouping of more than one hospital. If so, how many hospitals per HB?

Description of the subgroups examined should be much clearer. It looks like they have examined all hospital admissions, then performed subgroup analyses of those who were admitted to the ward (as opposed to ICU) with further stratification by therapy provided (oxygen and CPAP/HFNO). Standardised analyses should be provided and replicated for each subgroup.

Is the aim to compare outcomes by HB? If yes, include their outcomes in the paper along with the demographics of each group. If not (which I would recommend), I suggest removing this HB-specific information to an appendix and just reporting the overall numbers for all hospitals for each wave in the main manuscript.

HFNO and CPAP are not the same therapy, and require different degrees of intervention, monitoring, nursing attendance and technical maintenance. Is there data to differentiate the two? Was pressure support / bilevel non-invasive ventilation provided and recorded? Has this been aggregated with the CPAP group?

The authors also provide information for the outcomes for those admitted to ICU as a figure and report the mortality of those who receive CPAP/HFNO in ICU in the text. This is confusing to follow, particularly given they do not provide information about how many receive invasive ventilation, or how many went on from requiting CPAP/HFNO to needing ICU admission with or without CPAP/HFNO or invasive ventilation. How were patients classified if they required CPAP both in the ward and in ICU (latter presumably)? How many patients in each of these groups required invasive mechanical ventilation.

Results

The presentation of results is disorganised. I would suggest presenting the result for the whole study cohort, then the comparison of the waves for the whole study cohort, then reporting the subgroups (which need to be more clearly delineated in the Methods section).

More information should be provided about inclusions and exclusions. How were patients who were admitted to hospital more than once dealt with? Why were patients with LOS < 1 day excluded? Did this result in the exclusion of early deaths also? Where were the remaining 60% of the 14,614 patients? An inclusions/exclusions flow chart should be provided.

Missingness is provided for some variables in the supplement but not for many of the other variables reported in the main manuscript.

Is there information about how long patients spent in the hospital?

Line 133 – ‘datasets’ do the authors mean observations in their dataset or that there were actually 6444 datasets?

Table 1 – There is non-standard reporting of proportions – better as n (%). There is no table of overall outcomes. This should be provided for all patients, those in each wave and for other relevant subgroups.

The numbers went up in some HBs and down in others over the 3 waves. Is this a consequence of the waves affecting hospitals/regions differently or were there differences in data collection or missingness which affected this?

Mortality and numbers for those admitted to ICU are shown in figure 1, but I can’t see where the actual numbers of patients admitted to ICU are described. Given mortality is the main outcome measure, the actual values should also be provided in addition to the figure.

Fig 1 – ICU mortality - Clarify what this is: ‘in-ICU mortality’ or ‘in-hospital mortality of all patients admitted to ICU’ or ‘case fatality rate of any patient with COVID admitted to ICU’ etc

Fig 2 and 3 look almost exactly the same. While I recognise the authors state that multivariable adjustment had little influence on the results of univariate logistic regression, I am surprised by how little change there is. I can’t see any change for most variables. Can this please be checked? Could high levels of imputation have affected this?

Authors state frailty data was recorded but none is reported in the main paper. This would be an important risk adjustor to include in the main analysis if possible. Frailty completeness data is provided in the appendix but it is not clear if this is for all admissions, over 65s, nor what frailty score has been used and how this should be interpreted.

Table 1 is difficult to read and should be reformatted. See earlier comment about reporting by HB also, and consider removing this information from the table in the main manuscript.

Does figure 5 refer to CPAP alone or CPAP and HFNO? If the latter, then differences in the proportion of patients who received CPAP or HFNO between ICU and the ward might explain their results. This is before even considering the competing events of invasive ventilation (or death).

Do the authors have information on time spent receiving each of therapies?

Adjusted results are provided for CPAP in ICU vs ward, but no raw mortality. The full adjusted analysis should be provided in table form for any primary outcome assessed in the whole group or in a subgroup.

Have the authors accounted for between site variation in outcomes affecting their overall results, and how completeness of information at each site might have affected their study? E.g mixed effects modelling, site/HB as RE?

Discussion

The discussion might be better presented in a more standard way, where all results are summarized briefly at the beginning, followed by comparison to literature and then putting this together with implications of the study.

PCR testing – were pick-up diagnostic rates good during the first wave? Could cases have been missed and if so could these have affected your findings?

The authors state that a strength is that this was “… a truly national…” dataset. However information on 60% of COVID admissions is missing. Can this be truly nationally representative?

Having complete data for mandatory fields - is this a strength? This is exactly what I would expect for mandatory data fields.

The inclusion of community acquired COVID is also not a specific strength of the study. This is just their study group. Their statement makes it sound like they are pleased they have made their mortality numbers look better than they really are by excluding hospital acquired COVID. If truly reporting the mortality associated with COVID-19, then all cases should be included. Thus I think this is a limitation of their study, not a strength.

The authors mix comparisons with literature into their strengths and limitations sections.

Minor points

There are grammatical mistakes which need a thorough read through and check

Line 133 - ‘missing data imputation’ no capitalisation of M.

Line 147 – ‘Mortality rates and admission numbers for all patients was…’ should be ‘…were…’

Authors should avoid subjective language and use simpler statements which summarise data which can be read from the tables & figures. E.g. “There was a clear effect of age with significantly lower odds ratios in all ages before 70-79 and significantly higher odds ratios for ages from 80 onwards.” The finding would be the same irrespective of which reference group was used and could simply be stated with something like “Increasing age was associated with progressively higher mortality.”

Capitalisation of waves should be checked through for consistency whether using a nominative ‘Wave n’ or just describing waves (which is sometimes capitalised and sometimes not).

‘…on ICU’ – very English English from the Welsh! ‘…in ICU’ is standard terminology for the rest of the world. Please change.

COVID should be COVID-19 throughout the manuscript.

6. PLOS authors have the option to publish the peer review history of their article (what does this mean?). If published, this will include your full peer review and any attached files.

Reviewer #1: No

Reviewer #2: No

Reviewer #3: **Yes: **Dr. David Pilcher

---

## [Author Response · Author response to Decision Letter 0]

21 Jul 2023

These have been addresses and included in a table which has been attached

---

## [Decision Letter · Decision Letter 1]

30 Aug 2023

PONE-D-23-05105R1COVID-19 Managed on Respiratory Wards and Intensive Care Units: Results from the National COVID-19 Outcome Report in Wales From March 2020 to December 2021.PLOS ONE

Dear Dr. Barry,

Thank you for submitting your manuscript to PLOS ONE. After careful consideration, we feel that it has merit but does not fully meet PLOS ONE’s publication criteria as it currently stands. Therefore, we invite you to submit a revised version of the manuscript that addresses the points raised during the review process.

We look forward to receiving your revised manuscript.

Kind regards,

Academic Editor

PLOS ONE

Journal Requirements:

**Additional Editor Comments:**

Please revise.

Reviewers' comments:

Reviewer's Responses to Questions

**Comments to the Author**

1. If the authors have adequately addressed your comments raised in a previous round of review and you feel that this manuscript is now acceptable for publication, you may indicate that here to bypass the “Comments to the Author” section, enter your conflict of interest statement in the “Confidential to Editor” section, and submit your "Accept" recommendation.

Reviewer #1: All comments have been addressed

Reviewer #4: All comments have been addressed

2. Is the manuscript technically sound, and do the data support the conclusions?

Reviewer #1: Yes

Reviewer #4: Yes

3. Has the statistical analysis been performed appropriately and rigorously? 

Reviewer #1: Yes

Reviewer #4: Yes

4. Have the authors made all data underlying the findings in their manuscript fully available?

Reviewer #1: Yes

Reviewer #4: Yes

5. Is the manuscript presented in an intelligible fashion and written in standard English?

Reviewer #1: Yes

Reviewer #4: Yes

6. Review Comments to the Author

Reviewer #1: Dear authors, thank you for replying to all queries made re your mansucript. I have no further questions that require a response from the authors.

Reviewer #4: This is an epidemiology study of COVID-19 cases in 3 waves across Wales. The corrections were done extensively in revision 1.

Lines 272-280 on discussing the revolution of COVID-19 treatment modalities in all 3 waves needed a more thorough explanation. A timeline on which the specific treatment was initiated should be included.

Please include the specific time when vaccination was introduced.

Please provide an insight into how the expansion of knowledge and revolution of treatment help reduce the total length of hospital stay and the overall medical cost.

7. PLOS authors have the option to publish the peer review history of their article (what does this mean?). If published, this will include your full peer review and any attached files.

Reviewer #1: No

Reviewer #4: No

---

## [Author Response · Author response to Decision Letter 1]

18 Sep 2023

In response to reviewer 4, we have added a new figure (figure 5), indicating the admissions during each wave, mortality rates and a timeline of some of the key interventions

---

## [Decision Letter · Decision Letter 2]

13 Nov 2023

COVID-19 Managed on Respiratory Wards and Intensive Care Units: Results from the National COVID-19 Outcome Report in Wales From March 2020 to December 2021.

PONE-D-23-05105R2

Dear Dr. Barry,

We’re pleased to inform you that your manuscript has been judged scientifically suitable for publication and will be formally accepted for publication once it meets all outstanding technical requirements.

Kind regards,

Chiara Lazzeri

Academic Editor

PLOS ONE

Additional Editor Comments (optional):

Reviewers' comments:

Reviewer's Responses to Questions

**Comments to the Author**

1. If the authors have adequately addressed your comments raised in a previous round of review and you feel that this manuscript is now acceptable for publication, you may indicate that here to bypass the “Comments to the Author” section, enter your conflict of interest statement in the “Confidential to Editor” section, and submit your "Accept" recommendation.

Reviewer #1: All comments have been addressed

2. Is the manuscript technically sound, and do the data support the conclusions?

Reviewer #1: Yes

3. Has the statistical analysis been performed appropriately and rigorously? 

Reviewer #1: Yes

4. Have the authors made all data underlying the findings in their manuscript fully available?

Reviewer #1: Yes

5. Is the manuscript presented in an intelligible fashion and written in standard English?

Reviewer #1: Yes

6. Review Comments to the Author

Reviewer #1: Dear Authors,

Thank you for adressing my comments. I have no further queries regarding this manuscript.

7. PLOS authors have the option to publish the peer review history of their article (what does this mean?). If published, this will include your full peer review and any attached files.

Reviewer #1: No

---

## [Editor Report · Acceptance letter]

28 Nov 2023

PONE-D-23-05105R2 

COVID-19 Managed on respiratory wards and intensive care units: Results from the national COVID-19 outcome report in wales from March 2020 to December 2021. 

Dear Dr. Barry:

I'm pleased to inform you that your manuscript has been deemed suitable for publication in PLOS ONE. Congratulations! Your manuscript is now with our production department. 

Kind regards, 

on behalf of

Dr. Chiara Lazzeri 

Academic Editor

PLOS ONE